# A Brief Overview of the Rapid Progress and Proposed Improvements in Gallium Nitride Epitaxy and Process for Third-Generation Semiconductors with Wide Bandgap

**DOI:** 10.3390/mi14040764

**Published:** 2023-03-29

**Authors:** An-Chen Liu, Yung-Yu Lai, Hsin-Chu Chen, An-Ping Chiu, Hao-Chung Kuo

**Affiliations:** 1Department of Photonics, Institute of Electro-Optical Engineering, College of Electrical and Computer Engineering, National Yang Ming Chiao Tung University, Hsinchu 30010, Taiwan; 2Research Center for Applied Sciences, Academia Sinica, Taipei 114699, Taiwan; 3Institute of Advanced Semiconductor Packaging and Testing, College of Semiconductor and Advanced Technology Research, National Sun Yat-sen University, Kaohsiung 804201, Taiwan; 4Semiconductor Research Center, Hon Hai Research Institute, Taipei 114699, Taiwan

**Keywords:** gallium nitride, high electron mobility transistor, GaN on Si/SiC/QST

## Abstract

In this paper, we will discuss the rapid progress of third-generation semiconductors with wide bandgap, with a special focus on the gallium nitride (GaN) on silicon (Si). This architecture has high mass-production potential due to its low cost, larger size, and compatibility with CMOS-fab processes. As a result, several improvements have been proposed in terms of epitaxy structure and high electron mobility transistor (HEMT) process, particularly in the enhancement mode (E-mode). IMEC has made significant strides using a 200 mm 8-inch Qromis Substrate Technology (QST^®^) substrate for breakdown voltage to achieve 650 V in 2020, which was further improved to 1200 V by superlattice and carbon-doped in 2022. In 2016, IMEC adopted VEECO metal-organic chemical vapor deposition (MOCVD) for GaN on Si HEMT epitaxy structure and the process by implementing a three-layer field plate to improve dynamic on-resistance (R_ON_). In 2019, Panasonic HD-GITs plus field version was utilized to effectively improve dynamic R_ON_. Both reliability and dynamic RON have been enhanced by these improvements.

## 1. Introduction

GaN was a wide-bandgap semiconductor material with excellent electrical and optical properties [1,2,3], making it a promising candidate for various electronic and optoelectronic devices. In particular, the unique properties of GaN make it a popular choice for high-power and high-frequency applications [4,5,6,7], such as power electronics [8,9], RF amplifiers [6,7], and light-emitting diodes (LEDs) [10,11,12,13]. One of the key challenges in realizing the full potential of GaN devices was the quality of the epitaxial layer. The epitaxial layer was a thin layer of GaN that was grown on a substrate, and it plays a critical role in determining the performance and reliability of GaN devices [14,15,16,17,18,19]. Therefore, it was essential to have a thorough understanding of the epitaxial growth process and the structural properties of the epitaxial layer [20,21,22,23,24,25].

In recent years, significant progress has been made in improving the quality of GaN epitaxial layers through the development of advanced growth techniques and optimization of process parameters. The problem in GaN epitaxial layers was attributed to several factors, including defects [26,27], traps [28], and dislocations [29] in the material. These defects acted as sites for electron trapping and recombination, leading to localized heating and thermal runaway [30,31,32]. In addition, the high electric fields in GaN devices led to impact ionization and the creation of electron-hole pairs, which further deteriorate the breakdown problem [33]. To address the breakdown problem in GaN devices, scientists developed various solutions. One approach was to optimize the growth conditions of GaN epitaxial layers to reduce the density of defects and dislocations. This was achieved by using advanced growth techniques, such as MOCVD and molecular beam epitaxy (MBE), and by incorporating buffer layers and strain engineering [34,35,36]. Another approach was to develop new device structures and architectures that could mitigate the breakdown problem. For example, the use of field plates [37,38], edge termination structures [39,40], and deep trench isolation [41] helped to reduce the electric field concentration and improve the breakdown voltage of GaN devices. In addition, the use of advanced gate dielectrics, such as Al_2_O_3_ and HfO_2_, helped to reduce the gate leakage and improve the device’s reliability [42,43,44].

Overall, the breakdown problem in GaN epitaxial layers was a complex issue that required a multidisciplinary approach involving materials science, device physics, and electrical characterization. By developing advanced growth techniques and device structures, and by improving the understanding of the underlying physics, scientists were able to overcome the challenges and unlock the full potential of GaN devices for high-power and high-frequency applications. To improve these challenges, various characterization techniques have been developed to evaluate the structural properties of GaN epitaxial layers. These techniques include optical microscopy (OM) [45], atomic force microscopy (AFM) [46], photoluminescence (PL) [47,48], X-ray diffraction (XRD) [49,50], and transmission electron microscopy (TEM) [51,52], among others. These techniques enable researchers to evaluate the structural properties of GaN epitaxial layers at different scales and provide valuable insights into the growth process and the quality of the epitaxial layer.

In this paper, we review the recent advances in GaN epitaxial growth and the characterization techniques used to evaluate the structural properties of the epitaxial layer. We also discuss the challenges that need to be improved and the future directions for the development of GaN devices. The goal of the study is to provide a comprehensive overview of the current state of the art in GaN epitaxy and research in the enhancement of GaN devices. With the increasing issue of global warming, the demand for energy-efficient devices with a reduction in carbon increases. The development of third-generation semiconductors, including GaN on Si [53,54,55,56,57,58] or silicon carbide (SiC) [28,59,60,61,62,63], has been a focus of research for more than 20 years. Among these materials, GaN on Si has emerged as a promising alternative to SiC due to its lower cost and CMOS process compatibility [64,65,66,67,68,69]. Despite this advantage, the epitaxy of GaN on Si remains challenging due to the lattice mismatch and thermal expansion coefficient differences between the two materials.

Nonetheless, a few companies, such as TSMC [70,71] and Panasonic [72,73,74,75], have successfully produced GaN on Si, while other companies, such as IMEC, have made significant progress in device performance by focusing on the device designs, such as integrating the optimized material for epi-stack [76]. Recently, IMEC reported achieving a device of GaN on Si that reached 1200 V [77,78,79], with performance comparable to that of SiC. The fabrication process involved the use of a CMOS-fab process-friendly method in 2022 [80,81]. However, it is worth noting that the production of SiC substrates remains a high-energy-consuming industry. The crystal growth furnace requires temperatures exceeding 2500 °C [76,82,83,84,85,86], and the epitaxy process demands temperatures exceeding 1500 °C [87,88,89,90,91,92,93].

Given these challenges, the development of more sustainable and efficient production processes for GaN on Si and other third-generation semiconductors becomes the most preferred task for the industry. This would help to further reduce the environmental impact of the semiconductor industry while also enhancing the economic viability of these materials. Ultimately, GaN on Si and SiC represent important developments in the field of semiconductors, offering a pathway to more energy-efficient and environmentally friendly electronic devices [94,95,96,97,98].

## 2. GaN Epitaxy

GaN epitaxy on Si (111) faces a problem related to the lattice mismatch between the substrate and the epitaxial layers [99,100,101,102,103,104,105,106,107,108,109]. This mismatch can result in a high density of threading dislocations and other defects, which can affect the performance of the final device. To mitigate this issue, various techniques, such as strain engineering, defect reduction, and the use of graded buffer layers, have been developed to reduce the number of dislocations and improve the quality of the epitaxial layers. Moreover, the thermal management of the GaN on the Si substrate was another critical factor that needs to be considered during epitaxial growth. The high thermal conductivity of Si can cause significant thermal stress on the GaN epitaxial layer, which can result in cracking and other defects. Therefore, appropriate thermal management techniques such as the use of thermal interface materials, optimized structure design, and the integration of heat sinks need to be implemented to ensure the reliability and long-term performance of the device.

The Qromis Substrate Technology (QST) substrate structure was designed specifically for epitaxial growth and was characterized by a coefficient of thermal expansion (CTE) that was closely matched to the epitaxial layers grown on it. The substrate was composed of several layers, including a polycrystalline ceramic core that provided structural support, a first adhesion layer coupled to the core, a conductive layer coupled to the first adhesion layer, a second adhesion layer coupled to the conductive layer, and a barrier layer coupled to the second adhesion layer. In addition to these layers, the substrate also included a silicon oxide layer coupled to the support structure, which was then followed by a substantially single crystalline silicon layer. Finally, an epitaxial III–V layer was coupled to the single crystalline silicon layer. Overall, the QST substrate structure was designed to provide a stable and reliable foundation for epitaxial growth while also allowing for the use of a variety of semiconductor processing techniques [110]. An example of this would be the growth of thicker buffer layers on QST substrates [100,111,112], which requires the introduction of tensile stress to limit the increase in in-situ curvature. Figure 1 illustrates the improvement in curvature by introducing the alternate compressive and tensile stresses. QST substrates could be used for growing epitaxial thin films of various materials, such as nitrides, phosphides, arsenides, and other semiconductor materials. The characteristic of QST substrates was that their surfaces were very smooth with low defect density, which could provide high-quality surfaces that facilitated the growth and manufacturing of epitaxial thin films. In addition, the lattice matching of QST substrates could improve the crystal quality and lattice defect density of epitaxial thin films. QST substrates also had high transparency and optical performance, making them an ideal substrate for manufacturing high-quality optoelectronic devices.

To overcome the challenges posed by the mismatch in CTE between Si and (Al)GaN, which affected the mechanical strength of wafers during the epitaxial growth of buffers, additional measures were taken. High-quality (Al)GaN buffers could be grown on Si substrates, but the CTE mismatch put a limit on the buffer quality, thickness, and substrate size. One approach that was tried was to use substrates with nonstandard thickness, such as 1150 µm in our GaN-on-Si technology, for GaN growth on 200 mm Si substrates. However, this was not a scalable solution. A more practical solution was to use commercially available SEMI standard thickness engineered substrates, such as QST^®^, which had a CTE-matched poly-AIN core to the (Al)GaN buffer and good thermal conductivity. These substrates demonstrated extraordinary properties, including high crystal quality, abundant buffer thickness (>15 μm), high thermal conductivity, and scalability potential to large diameter (12 inches). By using such substrates, the limitations posed by the CTE mismatch and the mechanical strength issues during epitaxial growth of buffers were overcome, enabling high-quality (Al)GaN growth on Si substrates with improved mechanical and thermal properties. In summary, QST substrates are high-quality, high-performance substrates used for growing epitaxial thin films of various materials, including nitrides, phosphides, arsenides, and other semiconductor materials. Figure 1 illustrates that this can be accomplished by introducing alternate compressive and tensile stresses. However, this alone was not enough, as the strain partition effect will continue to be the primary impediment to the growth of thicker buffers for high-voltage applications. Traditional buffer technology that uses stepped superlattices [113,114,115,116] (SLs)—specifically, SL-1 > SL-2 (as indicated by the blue line in Figure 1)—relies on the gradual accumulation of compressive stress, which causes significant curvature in the QST substrates during growth. As a result, to achieve the growth of thicker buffers on QST substrates, the introduction of tensile stress was necessary to restrict in situ curvature buildup. This can be accomplished by alternating compressive and tensile stress in SL-1 < SL-2 (as shown by the purple line in Figure 1). Additionally, a reverse SL can be stacked to optimize the ex-situ wafer bow.

Furthermore, the interface between the GaN epitaxial layer and the Si substrate can also affect the performance of the device. The interfacial properties, such as the interface roughness, oxide layer thickness, and defect density, can have a significant impact on the electrical and optical properties of the device. Thus, the optimization of the interfacial properties is critical for achieving high-performance GaN on Si devices. In summary, the epitaxial growth of GaN on Si faces various challenges related to buffer layer design, defects, thermal management, and interfacial properties. Solving these challenges requires careful consideration of multiple factors and the implementation of appropriate techniques to ensure high-quality devices with reliable performance.

After the epitaxy process, the quality of the epitaxial layer was evaluated through the measurements of PL, XRD, AFM, and surface scans (surfscan). The information obtained from these measurements was used to optimize the epitaxial structure and refine the MOCVD recipe for subsequent epitaxy runs [117,118,119,120]. If any surface defect cracks were found on the wafer, further investigations were carried out to identify the cause and improve the quality of the epitaxial layer.

To improve the issue of gallium on Si substrate in contact resulting in melt-back etching, a nucleation layer of GaN or AlN can be adopted as an interface layer. However, due to limitations in the growth process, only an AlN nucleation layer (NL) can be selected in practice. The surface morphology and presence of unintentional oxygen impurities govern the vertical leakage of AlN NL/Si [121,122,123,124,125,126]. Interestingly, the AlN NL influences the growth of subsequent epitaxial layers as well as their vertical breakdown voltage. Further, it was found that the AlGaN intermediate layer and multi-pairs of AlGaN/AlN strained layer superlattice grown over AlN NL with better surface properties enhance the vertical breakdown voltage. Before applying the AlN nucleation layer, surface treatments such as spraying some aluminum or NH_3_ can be employed to create a rough SiN_x_ surface. This helps to militate the lattice mismatch-induced stress between the Si substrate and GaN lattice, which could otherwise lead to cracks and curvature in the epitaxial layer.

To further mitigate this issue, graduated buffer layers with a decreasing aluminum concentration were used, and a stepped gradient layer of Al_x_Ga_1−x_N is typically used instead of a linear gradient layer. Additionally, superlattice or interrupted layers of AlN can be inserted to enhance the crystal quality of the epitaxial layer. The carbon-GaN layer was also critical for the performance of the device due to its high insulator characteristic, as it determines the breakdown voltage and leakage [127]. Therefore, achieving uniformity in carbon doping was essential. Several options for carbon doping were available, including CH_4_, C_2_H_4_, C_3_H_8_, and CBr_4_. Some of the literature also suggests the use of Fe doping, which can improve the electrical conductivity of the epitaxial layer.

In 2020, IMEC planned to utilize 200 mm 8-inch QST substrate technology to grow GaN on silicon [128,129], which is a promising development for the electronics industry. This technology offers several benefits over traditional substrates, including reduced parasitic effects, a thermal expansion coefficient of AlN that matches the substrate, high thermal conductivity, high mechanical yield, and the capability to grow thick GaN buffer layers. These advantages enable the achievement of a high breakdown voltage of 650 V, which is essential for high-power devices.

To achieve this breakdown voltage, a layer of 200 nm AlN is grown at high temperatures on QST, followed by the growth of a 3900 nm superlattice structure, as shown in Figure 2, at a lower temperature. This structure functions as a template for the subsequent growth of GaN layers. Next, 1000 nm carbon-doped GaN is grown at the same temperature, followed by 400 nm of GaN. Finally, 12.5 nm of aluminum nitride was grown at high temperatures, and 80 nm of Mg-doped GaN was grown by cooling. In the context of growing Mg-doped GaN by cooling typically meant that the growth process involved a technique known as epitaxial lateral overgrowth (ELO) or lateral epitaxial overgrowth (LEO). During this process, a thin layer of GaN was first deposited on a substrate. Then, a mask was applied to the surface of the GaN layer, leaving only small openings for the growth of GaN crystals. When additional GaN material was deposited over the mask, the crystals that grew through the openings started to merge and form a continuous layer. The key aspect of ELO/LEO was that during the growth process, the temperature was lowered to a point where GaN crystals could only grow laterally over the substrate rather than growing vertically. This allowed for a more controlled and precise growth of the Mg-doped GaN layer, as it prevented the formation of defects and impurities that could arise during the traditional vertical growth process. Therefore, by cooling in this context referred to the use of ELO/LEO to grow a high-quality and uniform layer of Mg-doped GaN by controlling the temperature during the growth process. This sequential layer growth process enables the creation of a high-quality and high-performance GaN on Si HEMT.

The high voltage power HEMT market demand electrical characteristic of the resulting device was impressive, with a breakdown voltage greater than 650 V and a low leakage current of 10 μA/mm^2^ for GaN HEMT with 36 mm of gate width (W_G_) and 16 μm of gate-to-drain distance [128]. Figure 2 device has a high threshold voltage of about 3.1 V and a low off-state drain leakage of less than 1 μA/mm. The horizontal trench isolation breakdown voltage exceeds 650 V, which indicates the high reliability and robustness of the device. Additionally, the device dispersion was well controlled to within 20% over high and low temperature and bias voltage ranges, which ensures consistent performance over a wide range of operating conditions. Overall, the use of QST technology to grow GaN on Si holds great promise for the development of more efficient and high-performance electronic devices. The combination of high breakdown voltage, low leakage current, and well-controlled dispersion makes this technology an attractive option for power electronics, high-speed communications, and other demanding applications [130].

In recent years, the development of high-breakdown voltage epitaxial structures has been a major focus in the field of GaN power electronics. In 2020, IMEC achieved a significant milestone by developing an epitaxial structure with a breakdown voltage of 650 V. Building on this success, in 2022, they were able to increase the breakdown voltage to 1200 V by using the 200 mm QST substrate, showcasing the potential of complex epitaxial material stacks and QST substrates for high-voltage power applications such as electric vehicles. To achieve high breakdown voltage, the coefficient of thermal expansion (CTE) of the poly-AlN substrate is a crucial factor [81,131]. IMEC has carefully designed the poly-AlN substrate to closely match the CTE of the GaN/AlGaN epitaxial layer, enabling thicker epitaxial structures to be grown on large diameter substrates while maintaining the mechanical strength of the substrate and achieving higher voltage operation.

Besides the structural improvements, IMEC has also made significant advancements in its MOCVD technology. They have replaced the vertical Veeco MOCVD with the AIXTRON horizontal MOCVD [132,133], leading to improved epitaxial structures, as shown in Figure 3. IMEC has also experimented with different epitaxial structures using reversed stepped superlattice (RSSL) inverted stepped superlattice structures, achieving higher vertical breakdown voltage >1200, as shown in Figure 4a–c. IMECs research and development efforts have made significant contributions to the field of GaN power electronics. Furthermore, they have collaborated with leading semiconductor companies to develop commercial-grade GaN-based power electronics for various applications, which has further accelerated the adoption of GaN-based power electronics in the market. The success of IMECs research and development in GaN-based power electronics is expected to pave the way for the next generation of high-efficiency and high-power electronic systems.

The breakdown voltage of GaN HEMT was an important parameter that determined their reliability and performance. The critical electric field of AlN was higher than that of GaN, making it necessary to increase the Al percentage in the superlattice region to achieve high breakdown voltage. Additionally, the optimization of carbon concentration was crucial in achieving high breakdown voltage. Carbon doping can be performed through external sources such as ethylene or by tuning epitaxy parameters that affect crystal quality [134,135,136].

In this chapter, we also explore another way to increase breakdown voltage, which was to reduce the critical electric field at the AlN/Si interface by increasing the thickness of the buffer layer. There were different structural concepts and optimized Figure 4a–c to achieve a target vertical breakdown voltage of at least 1200 V. Design grow stress relief layers with varying thicknesses for structures A, B, and C and evaluate their breakdown voltage under high and low-temperature conditions [81].

Results demonstrate that the epitaxial structure plays a significant role in determining the collapse voltage of GaN HEMT. The doping of carbon was the main factor that affected breakdown voltage. It found that the breakdown voltage of structures A and C exceeds 1200 V, while structure B did not reach 1200 V for both thicknesses tested. Moreover, the thickness of structure C was less than that of structure A, but its breakdown voltage at high and low temperatures exceeds 1300 V. These results emphasize the importance of optimizing the epitaxy structure and carbon doping concentration for achieving high breakdown voltage in the GaN HEMT.

In the field of epitaxial growth of GaN, it is essential to ensure the quality of the surface since it affects the subsequent processes and the final characteristics of the components. Therefore, the surface inspection was critical in ensuring the quality of the epitaxial layer. OM was commonly used to check the surface of the substrate and epitaxial layer for visible cracks and roughness. AFM was a high-resolution technique used to detect surface roughness and morphology at the nanoscale level. PL was a non-destructive technique used to evaluate the quality of the epitaxial layer by measuring its optical properties. XRD is a powerful technique that can provide information about the crystal quality, thickness, and strain of the epitaxial layer.

In addition to these techniques, TEM can also be used to investigate the structural defects in the epitaxial layer at the atomic level. It can provide detailed information about the crystal structure, defects, and dislocations, which are critical for determining the quality of the epitaxial layer. Apart from the above-mentioned techniques, KLA Corporation’s product of surface scan is another widely used method to detect surface defects and particles [137]. It was a non-contact, high-resolution technique that can detect surface defects such as scratches, pits, and particles that are smaller than the wavelength of light. In conclusion, the inspection of the surface quality of the epitaxial layer is crucial for ensuring the quality of the final components. The use of various surface inspection techniques such as OM, AFM, PL, XRD, TEM, and KLA Corporation’s surface scan provides critical information about the quality of the epitaxial layer and helps in improving the process and component performance. Three typical substrate architectures in Figure 5a have been individually optimized for epi-quality and electrical performance. A step-graded buffer uses, on top of an AlN nucleation layer, several AlGaN layers with decreasing Al content in steps. A superlattice buffer Figure 5b consists of N pairs of strained AlN/(Al)GaN layers. This allows for the growth of thicker buffers while keeping the overall wafer bow below 50 μm. Increasing the (Al)GaN buffer thickness increases the achievable breakdown voltage. Increasing the thickness of the GaN buffer using a step-graded approach, however, increases the in-film stress resulting in a high density of slip lines, high wafer bow, and possibly film and wafer cracking. Therefore, thin low-temperature AlN interlayers Figure 5c can be inserted in a step-graded type of buffer to minimize the internal stress during growth and post-growth cool down, thus achieving thicker layers [138].

## 3. GaN Process

On the other hand, depletion-mode (D-mode) devices have been developed as an alternative solution to overcome the limitations of power switching capability in Si-related devices. D-mode devices can be fabricated using a self-aligned process and offer several advantages, including a simple fabrication process, low off-state leakage current, and high stability. These advantages made D-mode devices suitable for power-switching applications, particularly for fast switching speed and high-current operations. In recent years, significant progress has been made in the development of GaN-based enhancement-mode (E-mode) and D-mode devices, particularly in terms of their performance, reliability, and scalability. However, several challenges still need to be improved, including the optimization of the key material selection, device structure design, and process precise control. This article aims to provide an overview of the GaN process’ current state, including the remaining challenges and opportunities for GaN development and the future prospects for GaN-based E-mode and D-mode devices, focusing on their performance, reliability, and scalability [139,140,141].

One of the key features of GaN-based devices was their ability to operate in the D-mode, which was attributed to the unique material properties of the two-dimensional electron gas (2DEG) in AlGaN/GaN. Furthermore, E-mode devices have been extensively studied and developed for their ability to offer high performance, low R_ON_, and low dynamic power consumption. However, the fabrication of E-mode devices requires precise control of the doping process in the gate structure, which has several challenges, such as the high gate leakage current and poor stability under high temperature and high voltage conditions.

An AlGaN/GaN of E-mode and D-mode HEMT combined inverter, as shown in Figure 6, utilizing p-GaN gate technology on a 150 mm silicon wafer. The key difference between an E/D mode GaN inverter and other conventional inverters lies in the use of a p-GaN epitaxial layer on a 150 mm silicon substrate, which was fabricated on a p-GaN gate/AlGaN/GaN. The epitaxial layer is grown using the MOCVD technique. It includes a 200 nm AlN nucleation layer, a high-resistance buffer layer of about 3.5 µm, an undoped GaN channel layer of about 300 nm, and an Al_0.13_Ga_0.87_N barrier layer of about 18 nm. These layers were sequentially grown on the silicon substrate to form the epitaxial structure. To complete the fabrication process, a p-type GaN layer of about 90 nm was grown with an Mg concentration of about 10^19^ cm^−3^. The entire structure was then annealed at 650 °C for 15 min in the MOCVD. This process helps to activate the Mg dopants and enhance the electrical properties of the device [142].

After the epitaxial growth and annealing, the process continues with device fabrication steps. First, mesa isolation was performed using standard photolithography and dry etching techniques [143,144]. After performing mesa isolation using standard photolithography and dry etching techniques, surface leakage problems may have occurred. Surface leakage refers to the phenomenon where current flowed along the surface of the mesa, rather than through the desired pathway. This could have led to reduced device performance or failure. One possible cause of surface leakage was the presence of residual etching material on the mesa surface which could have created a conductive path. This could have been mitigated by thoroughly cleaning the mesa surface after etching, using techniques such as plasma cleaning or wet chemical cleaning. Another possible cause of surface leakage was the formation of surface defects during the etching process. These defects could have acted as trap sites for charge carriers, allowing them to bypass the intended pathway. To minimize surface defects, it was important to optimize the etching conditions, such as gas composition, pressure, and power. In addition, proper mesa passivation could have also helped to reduce surface leakage. Passivation referred to the process of coating the mesa surface with a thin layer of insulating material, such as silicon dioxide or silicon nitride. This layer could have helped to block surface currents and reduce the likelihood of surface defects. Overall, careful attention to cleaning, etching conditions, and passivation could have helped to minimize surface leakage problems after dry etching mesa isolation. Then, the ohmic contacts were formed by depositing a Ti/Al/Ni/Au metal stack [145,146,147] and annealing at 850 °C for 30 s [148,149,150]. Next, the p-GaN gate was formed by depositing the Ni/Au metal stack and annealing at 500–800 °C [151,152,153]. Finally, the passivation layer was deposited to protect the device from environmental affecting. In the summary, the AlGaN/GaN E/D mode HEMT combined inverter using p-GaN gate technology on a 150 mm silicon wafer was successfully fabricated. The process involves the growth of the p-GaN epitaxial layer on a 150 mm silicon substrate, followed by device fabrication steps, including mesa isolation, ohmic contact formation, p-GaN gate formation, and passivation. The resulting device exhibits excellent electrical properties and can be utilized in various applications, such as power electronics and high-frequency circuit-based devices [154].

## 4. Gold-Free COMS-Compatible GaN Technology

The compatibility of GaN technology with complementary metal-oxide-semiconductor (CMOS) processes has been a long-standing challenge in the semiconductor industry. The demand for high-speed, high-power, and high-temperature electronic devices has driven the development of GaN-based technology. However, the integration of GaN devices with conventional CMOS processes has been limited due to the incompatibility of the materials and fabrication processes. To improve this issue, researchers have developed various CMOS-compatible GaN fabrication methods, including the use of Si substrates and Si-based materials. The use of Si substrates has several advantages, including low cost, large area, and well-established CMOS processing techniques. However, the lattice mismatch between Si and GaN results in high defect density and poor crystalline quality, which leads to low device performance [64,65,66,67,68,69].

To overcome these challenges, researchers have developed new techniques, such as the epitaxial lateral overgrowth (ELOG) of GaN on Si substrate. ELOG is a promising technique that allows the growth of high-quality GaN thin film on Si substrate by reducing the defect density and enhancing the crystalline quality of the GaN thin film. Another promising approach was the use of a strained Si substrate, which can reduce the lattice mismatch and improve the GaN crystal quality. Moreover, the development of CMOS-compatible GaN-based technology also involves the optimization of the process steps, such as the growth of high-quality GaN thin film, the formation of ohmic contact, and the integration of GaN devices with CMOS circuits. These steps require careful control of the growth conditions, surface treatment, and device processing, as well as the development of new materials and fabrication techniques.

In recent years, significant progress has been made in CMOS-compatible GaN technology, with several successful demonstrations of GaN-based power devices integrated with CMOS circuits. These devices have shown promising performance, including high breakdown voltage, low R_ON_, and fast switching speed. The development of CMOS-compatible GaN technology was expected to enable the integration of high-performance GaN devices with CMOS circuits for a wide range of applications, such as power electronics, RF communications, and optoelectronics.

This section focuses on the fabrication of a nitride-based semiconductor device using a CMOS-compatible process. The process was performed using a 200 mm CMOS-compatible process from IMEC, as shown in Figure 7a. The process involves depositing the TiN layer on the p-GaN layer and then etching it using a dry etch. This was followed by a selective dry etch from the p-GaN layer to the AlGaN layer using BCl_3_/SF_6_. The gate was then defined by laterally etching the TiN metal layer. This technique significantly suppresses leakage through the gate sidewall and improves the gate reliability. Afterward, dielectric layers using Al_2_O_3_ and SiO_2_ were deposited in defined areas, and nitrogen ion implantation was performed for lateral isolation. The dielectric layer was then etched away in the gate area, and then the gate metal layer was deposited and patterned.

Following the gate treatment, the ohmic contact metal layer of the source and drain was plated using a metal stack composed of Ti, Al, Ti, and TiN. The metal stack was then annealed at a low temperature. Trench isolation was formed by etching the dielectric AlGaN buffer layer down to the Si (111) layer [121,155], stopping at the SiO_2_ buried oxide layer. Chemical mechanical polishing (CMP) with SiO_2_ filling was then performed [156,157,158]. Through etch to the substrate, rear metallization and SiN passivation protection layer are added to the device. Finally, the entire component manufacturing process is completed, as shown in Figure 7b. The CMOS-compatible process used in this paper provides a promising pathway for the integration of nitride-based devices with mainstream CMOS technology, enabling the development of new and advanced electronic devices.

## 5. The Defect of GaN HEMT Device

In this part, the focus is on the reliability and defects of GaN devices. As shown in the Figure 8a, the electric field formula of the E-mode element under high bias can lead to rapid shortening of the device life due to defects [159]. Therefore, in addition to improving the epitaxial design structure, there were several ways to enhance device reliability. Firstly, fluorine ion implantation can be used under the gate. Secondly, the metal-insulator-semiconductor (MIS) type gate design can be employed. Thirdly, the high voltage GaN HEMT can be connected to the low voltage silicon MOSFET. Fourthly, a p-GaN or p-AlGaN layer can be added to the AlGaN/GaN HEMT structure.

Figure 8b shows the time to failure (TTF) study by IMEC using QST substrates with a breakdown voltage of about 200 V [160]. However, no carbon doping was added to the buffer layer, leading to reduced reliability. In contrast, the 2020 Aixtron paper sponsored by Bayerische Motoren Werke (BMW) changed from the general carbon-doped stepped buffer layer to 85 pairs of the carbon-doped superlattice, as can be seen in Figure 8b. This change has effectively doubled the device’s lifetime. Therefore, the design of the epitaxial structure changes is critical not only for improving component characteristics but also for enhancing device reliability [161].

On reliability and defects of GaN technology presents various approaches to improve the performance and reliability of the components. One of the strategies discussed the use of is the Panasonic HD-GITs Plus field version [162], which has been shown to effectively improve dynamic R_ON_. In 2012, IMEC designed an 8-inch Si substrate with a gold-free CMOS component design, which had a breakdown voltage of about 300 V. This design used the three-layer field plate to improve dynamic R_ON_. Compared with components using a one-layer field plate and a three-layer field plate, the normalized on-resistance dropped from 3.7 to 1.7 on average. Not only can a well-designed source and gate field-plate configuration reduce the magnitude of the electric field between the drain and source, but it can also constrain the trapping region by regulating the location of the electric field maxima [163]. In 2018, Panasonic introduced the Hybrid-Drain-embedded GITs (HD-GITs), which used the previously mentioned MIS structure and had a breakdown voltage of up to 730 V and a drain current of up to 20 A [66].

Figure 9 illustrates the distribution of the electric field in the HD-GITs. The electroluminescence signal peak was divided into the following two regions: the drain-side edge of the gate and the drain. These were the channels where the electric field was the strongest. The HD-GITs have demonstrated no-current collapse up to 1 kV. The trapping mechanism responsible for the current collapse was investigated in GITs and HD-GITs, combining pulse measurements, transient investigations, and electroluminescence characteristics under off-state and half-on conditions. To improve current collapse, the p-type drain was introduced in HD-GITs to help inject holes in closed and half-open states. The results showed that the traps filled in the closed state and the half-open condition were the same, with an activation energy (*E*a = 0.8 eV) consistent with the displacement of carbon defects.

The dynamic R_ON_ variation under half-on conditions, such as pulsed and transient measurements, was found to be different between GITs and HD-GITs. The faster demolding speeds in HD-GITs led to improved pulse measurements during the current collapse and reduced drain edge (semiconducting luminescence under constant voltage stress) conditions. Overall, the Panasonic HD-GITs Plus field version has shown great promise in improving dynamic R_ON_ and reducing current collapse, thereby enhancing the reliability of gallium nitride technology [164].

In 2019, Panasonic conducted research on the electric field of different thicknesses of AlGaN in the E-mode. Figure 10a,b the results of this study [165]. The electric field distribution in the AlGaN layer can be improved by reducing the thickness of the AlGaN layer. This helps to reduce the electric field peak, which can improve the reliability and lifetime of the device. In addition, Panasonic also conducted research on the HD-GITs (Hybrid-Drain-embedded GITs) plus field version in 2019. Figure 10c The HD-GITs plus field version can effectively improve the dynamic R_ON_ of the device [164]. This is achieved by introducing a drain layer in the HD-GITs, which helps to inject holes in closed and half-open states, and thus improves current collapse. The trapping mechanism responsible for the current collapse was also investigated in GITs and HD-GITs, which helps to understand and improve the reliability and lifetime of the device.

The research also found that the variation between GITs and HD-GITs is the dynamic R_ON_ variation under half-on conditions, such as pulsed and transient measurements. The faster demolding speeds in HD-GITs lead to improved pulse measurements during the current collapse and reduced drain edge conditions. Furthermore, the study found that the traps filled in the closed state and the half-open condition are the same, with activation energy consistent with the displacement of carbon defects. These findings provide valuable insights into the design and optimization of the epitaxial structure for improving the reliability and lifetime of GaN-based devices.

In 2015, Infineon conducted a study on the effect of measurement setup on pulsed I–V measurements, focusing on *I*_DS_(t) waveform to ensure the accuracy of the measurement. The study found that R_ON_ was insensitive to short transients, below 10 μs, in both the fresh and accelerated bias aging test (HTRB) aged devices. However, the R_ON_ doubled in shutdown when increasing V_DS0_ from 0 V to 50 V. Moreover, the threshold voltage V_TH_ of the aged device in the HTRB aging test, which was conducted for 2200 h, showed comparable trapping effects that led to an increase in R_ON_, particularly under the gate-source and drain-source access areas.

In 2015, On-semi and IMEC studied the use of ion implantation as an effective and controllable way to reduce the dynamic R_ON_. The effects were evaluated by combining pulse characterization, transient measurements, and electroluminescence analysis on untreated and irradiated devices. The study found that ion implantation led to an increase in electroluminescence signal and a decrease in dynamic R_ON_ post-implantation, which was attributed to the reduction of leakage in the u-GaN layer and the facilitation of charge release from the buffer layer. Notably, no trap level state (1.5 × 10^14^ ions/cm^2^) was detected after ion implantation. When a semiconductor material is implanted with ions, it can lead to the creation of defects in the lattice structure of the material, which can act as trapping centers for charge carriers. These trapping centers can affect the electrical properties of the material, such as its carrier mobility, lifetime, and resistance. In this case, the ion implantation was performed in such a way that the defects created in the lattice structure of the material did not act as trapping centers for charge carriers. This was likely achieved by optimizing the implantation conditions, such as the energy and dose of the implanted ions, to ensure that the created defects were either electrically neutral or did not significantly affect the electrical properties of the material. Furthermore, the study mentions that the ion implantation led to an increase in electroluminescence signal and a decrease in dynamic R_ON_, which suggests that the created defects did not have a significant impact on the material’s electrical properties. Therefore, it is reasonable to conclude that no trap level state was detected after ion implantation because the created defects did not act as trapping centers for charge carriers.

The characterization of low-frequency noise (LFN) in electronic devices at cryogenic temperatures became an active research area. For instance, researchers examined shot noise in short-channel MOSFETs, white noise in NMOS technology, and the microwave noise figure of InP HEMT under cryogenic conditions. The LFN level of devices was considered a crucial parameter in cryogenic electronic systems as it contributed significantly to the transistor’s comprehensive noise, especially given that both thermal and shot noise decreased at cryogenic temperatures. Additionally, LFN measurements could assess the defect states in devices and their reliability. Although various studies investigated the behavior and origins of LFN in Si-based and GaAs/InP-based cryogenic systems, few studies examined the LFN characteristics of GaN HEMTs in the cryogenic environment, particularly at extremely low temperatures down to 4.2 K.

To address this gap, we conducted a comprehensive study of the DC and LFN behavior of GaN-based HEMTs across a temperature range from 300 K to 4.2 K. As the temperature decreased, we observed an overall improvement in the devices’ electrical performance, with a 75% increase in maximum IDS (*I*_DS,max_), steeper SS, suppressed leakage current by an order of magnitude, and decreased R_ON_. These improvements were attributed to the suppressed lattice scattering and enhanced carrier mobility under cryogenic conditions. Additionally, the studies analyzed the LFN of the devices as a function of temperature, revealing a clear 1/f noise behavior. More importantly, we used the carrier number fluctuations with correlated mobility fluctuations (CNF/CMF) model to describe the origin and underlying physics of LFN down to 4.2 K. Research extracted and discussed the correlated parameters, i.e., flat-band voltage spectral density (Svfb) and trap density (Nt), which revealed the defect evolution in the GaN-based HEMTs under cryogenic temperatures. Overall, our work provided a thorough analysis of the DC and LFN characterizations of GaN HEMTs down to 4.2 K, offering insights for the design of future GaN-based cryo-devices and systems [166].

## 6. Conclusions

In conclusion, this review paper has highlighted the rapid progress and proposed improvements in GaN on Si epitaxy structure to achieve the high breakdown voltage and three-layer field plate to improve dynamic R_ON_. IMECs use of the 200 mm, 8-inch QST substrate with GaN epitaxy architecture for high breakdown voltage has achieved significant breakthroughs, with the breakdown voltage reaching 650 V in 2020 and 1200 V in 2022 through the use of superlattice and carbon-doping techniques. The adoption of VEECO MOCVD and the three-layer field plate has also improved the process in GaN HEMT, resulting in the low dynamic R_ON_. Furthermore, highlights a CMOS-fab compatible process, which provides a promising pathway for the integration of GaN devices with mainstream CMOS technology, thereby enabling the development of new and advanced electronic devices. Panasonic HD-GITs plus field version was also utilized to effectively improve dynamic R_ON_ in 2019. These improvements have led to enhanced reliability and R_ON_, marking a significant milestone in the development of third-generation semiconductors with wide bandgap, particularly GaN on Si.

## Figures and Tables

**Figure 1 micromachines-14-00764-f001:**
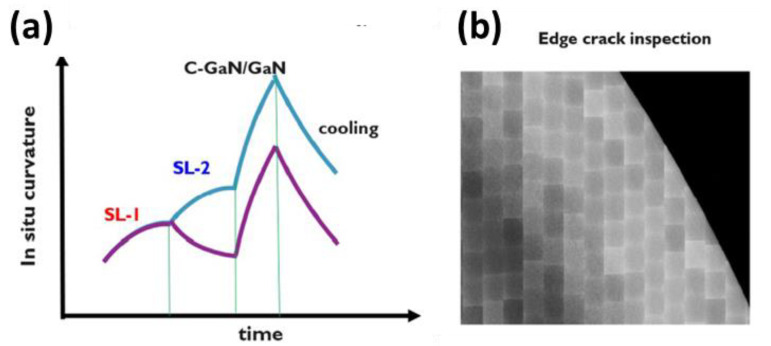
(**a**). Conceptual representation of stepped superlattice and reversed stepped superlattice. (**b**) CAMTEK edge inspection (the square pattern on the wafer is due to stitching of measurement shots) [81]. Figure reproduced with permission from AIP Publishing Applied Physics Letters.

**Figure 2 micromachines-14-00764-f002:**
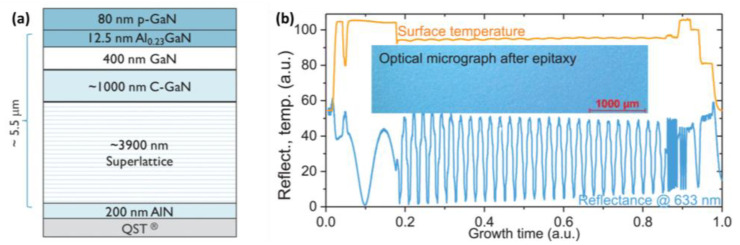
(**a**) Schematic cross-section of the epitaxial layer stack and (**b**) in-situ data of the surface temperature and the reflectance. The inset shows an optical micrograph of the surface after the epitaxial growth [128]. Figure reproduced with permission from IEEE Transactions on Semiconductor Manufacturing.

**Figure 3 micromachines-14-00764-f003:**
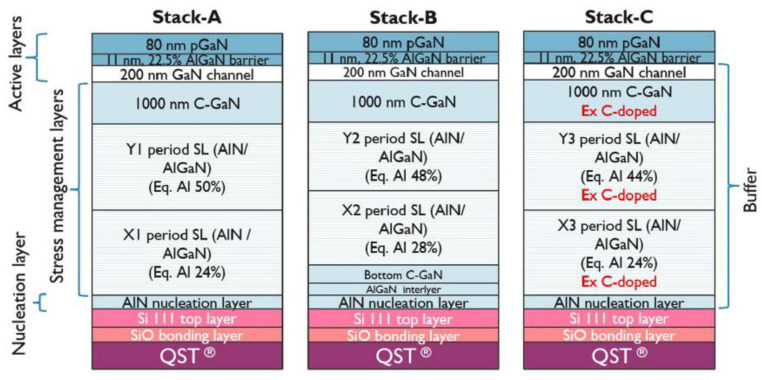
HEMT stacks based on intrinsic (stack A and stack B) and extrinsic (stack C) C-doping developed in this study. Buffer thicknesses of stack A and stack B vary from 5.3–7.4 to 4.8–6.1 μm, respectively. Stack C has a thickness of ∼6.8 μm [81]. Figure reproduced with permission from AIP Publishing Applied Physics Letters.

**Figure 4 micromachines-14-00764-f004:**
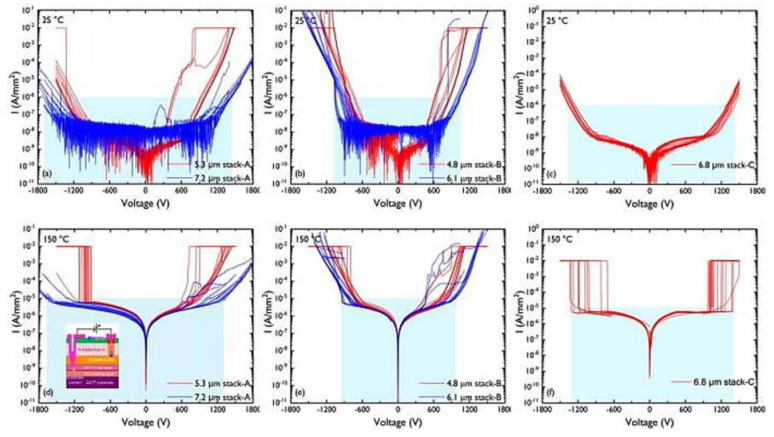
Vertical buffer leakage measurements for intrinsic C-doped (**a**,**d**) stack A (5.3 μm vs. 7.2 μm), (**b**,**e**) stack B (4.8 vs. 6.1 μm), and (**c**,**f**) stack C (6.8 μm) at 25 and 150 °C [81]. Figure reproduced with permission from AIP Publishing Applied Physics Letters.

**Figure 5 micromachines-14-00764-f005:**
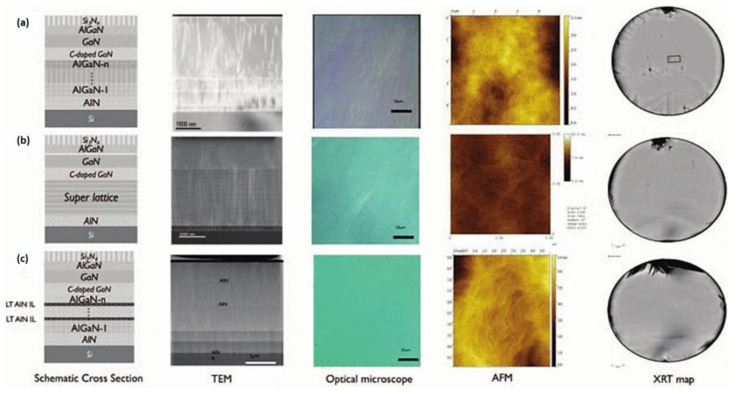
Schematic cross-section (not to scale), TEM, optical microscope, AFM, and X-ray topography map (XRT) of (**a**) stepped buffer, (**b**) superlattice, and (**c**) low-temperature AIN interlayer. Both the superlattice and interlayer approaches reduce the density of slip lines and cracks significantly with respect to the step-graded buffer approach [138]. Figure reproduced with permission from IEEE Proceedings.

**Figure 6 micromachines-14-00764-f006:**
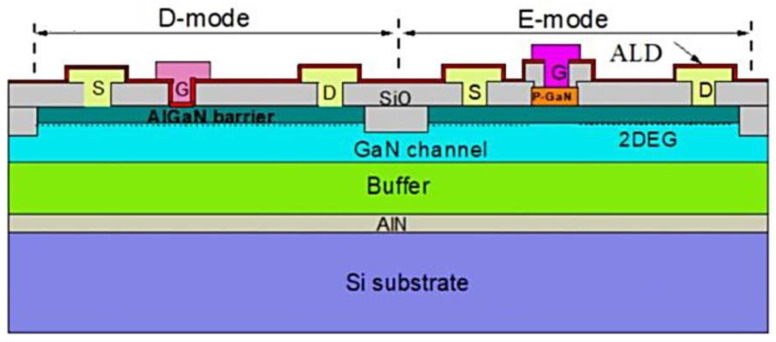
Schematic cross-section of E/D-mode GaN monolithic integration technology Adapted with permission from [142].

**Figure 7 micromachines-14-00764-f007:**
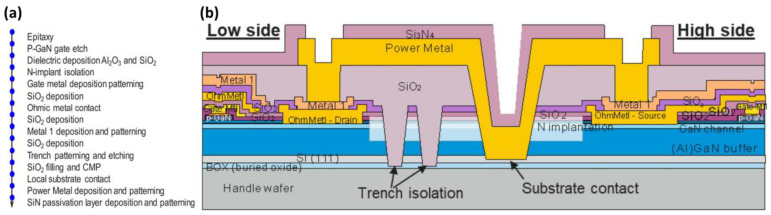
(**a**) Schematic cross-section of the epitaxial layer stack and process integration flow. (**b**) Schematic cross-section of the integrated half-bridge, trench isolation, and local substrate contact [128]. Figure reproduced with permission from IEEE Transactions on Semiconductor Manufacturing.

**Figure 8 micromachines-14-00764-f008:**
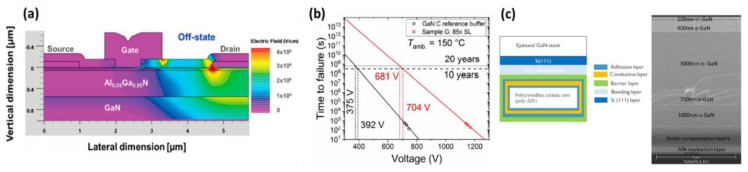
(**a**) A simulated electric field in a GaN high electron mobility transistor (HEMT) in off-state, with a drain bias of 300 V. (**b**) TTF (Weibull scale factor, 63.2% failure) dependence on the drain voltage along with lifetime extrapolations comparing the reference GaN:C and AlGaN/AlN superlattice buffer. (**c**) Schematic image of QST^®^ engineered substrates with poly-AlN core (with epitaxial GaN stack on top), and cross-sectional SEM image of a vertical device stack with a total GaN thickness of 5.4 μm on 200 mm engineered substrates [159,160,161]. Figure reproduced with permission from IEEE Transactions on Electron Devices and Elsevier Microelectronics Reliability.

**Figure 9 micromachines-14-00764-f009:**
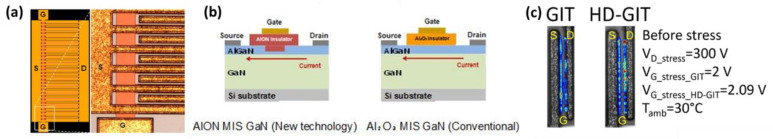
(**a**) Layout (left) and microscope picture detail (right) of a 60 mm gate width power device. (**b**) AlON MIS GaN (new tech.), and Al_2_O = MIS GaN (conventional). (**c**) A false color profile is taken after the application of *V*_D_ = 300 V, *V*_G__GIT = 2 V, and *V*_G__HD-GIT = 2.09 V. Under semi-ON conditions, the luminescence is localized at the drain side of the gate and at the drain contact both in the GIT and in the HD-GIT [66,162,164]. Figure reproduced with permission from IEEE Proceedings.

**Figure 10 micromachines-14-00764-f010:**
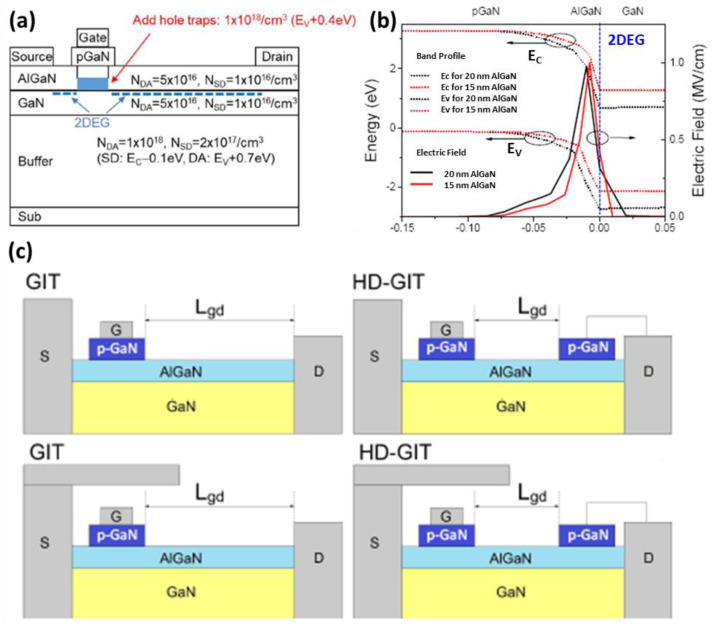
(**a**) Schematic cross-section of gate injection transistor (GIT). The densities and levels of traps employed in simulation in this study are also presented. (**b**) Energy band profiles underneath the pGaN gate simulated for the GIT with a thinner (15-nm-thick) AlGaN (larger *V*_TH_) and a thicker (20-nm-thick) AlGaN barrier (smaller *V*_TH_). (**c**) Schematic cross sections of a gate injection transistor (GIT) and a hybrid-drain-embedded GIT (HD-GIT) without and with the field plate [164,165]. Figure reproduced with permission from IEEE Proceedings.

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
