# Peer review of "A Brief Overview of the Rapid Progress and Proposed Improvements in Gallium Nitride Epitaxy and Process for Third-Generation Semiconductors with Wide Bandgap"

_micromachines, 2023, doi:10.3390/mi14040764_

Round 1

Reviewer 1 Report

The paper reviews progress in of GaN on Si devices focusing on quality issues. The topic is relevant nowadays and although there are a number of review papers on GaN devices, each of them focuses on different issues.

Some quality assessment techniques are overviewed. Low frequency noise measurement could also be discussed, as it is known as a sensitive method to assess quality and reliability of  semiconductor devices.

Formatting must be improved: now figures and their captures are on different pages.

English must be reviewed: style and grammar.

Author Response

Dear reviewer 1:

        Some quality assessment techniques are overviewed. Low frequency noise measurement could also be discussed, as it is known as a sensitive method to assess quality and reliability of semiconductor devices.

        Ans:Regarding the quality assessment techniques discussed in the paper, we will take your suggestion into consideration and include a section on low-frequency noise measurement as a sensitive method for assessing the reliability of semiconductor devices in lines 581-606.

        Formatting must be improved: now figures and their captures are on different pages.

        English must be reviewed: style and grammar.

        Ans:We have addressed the formatting issue by ensuring that figures and their captions appear on the same page. Additionally, we have carefully reviewed the English used in the paper to improve both style and grammar.

Reviewer 2 Report

After reading the paper entitled "A Review of the Rapid Progress and Proposed Improvements in Gallium Nitride Epitaxy and Process for Third-Generation Semiconductors with Wide Bandgap," several points need to be corrected for possible publication. Wide bandgap nitride-based power devices on silicon substrate are a vital area for today's and tomorrow's technologies. This "review" attempt is a challenging exercise to carry out, and it has been done partially in this work. However, it is not a complete review, and several corrections need to be made for more clarity on the subject.

To begin with, the title is not accurate. It is a reductionist view of the studies conducted on the subject and not a comprehensive review of the topic but rather partial. I suggest that the authors remove the word "review" from the title and replace it with a more specific term such as "brief overview" or something similar.

The problems encountered by the concerned community, namely technologists and people involved in electrical characterization, are not mentioned in the introduction and the beginning of this paper. Where do the breakdown problems in these components come from? What solutions are established by the scientists?

In the "GaN epitaxy" paragraph, please specify what QST layers of QROMIS are, provide the nature, thickness, and co-doping of the layers (wSiC, qqs um to qqs 10um, doped B or Al, etc.). Indicate whether oxide-type interface layers are used and their impacts on thermal conductivity. This lacks precision.

Not all growth studies on Si (100) or performance on Si (111) (100) (110) are cited. Here are some papers in the field that are not cited. For example:

1. Dual Role of 3C-SiC Interlayer on DC and RF Isolation of GaN/Si-based Devices

Abdelkrim El Hadi Khediri, Brahim Benbakhti, Jean-Claude Gerbedoen, Abdelatif Jouad, H. Maher, Nour Eddine Bourzgui and Ali Soltani, Applied Physics Letters, 121 (12)
doi.org/10.1063/5.0102644

2. Assessment of transistors based on GaN on silicon substrate in view of integration with silicon technology, A Soltani, Y Cordier, J-C Gerbedoen, S Joblot, E Okada, M Chmielowska, M R Ramdani and J-C De Jaeger
Semiconductor Science and Technology, Volume 28, Number 9, A Soltani et al 2013 Semicond.
Sci. Technol. 28, 094003. DOI 10.1088/0268-1242/28/9/094003

3. Assessment of transistors based on GaN on silicon substrate in view of integration with silicon technology, SOLTANI A., CORDIER Y., GERBEDOEN J.C., JOBLOT S., OKADA E., CHMIELOWSKA M., RAMDANI M.R., DE JAEGER J.C. , Semicond. Sci. Technol., 28, 9 (2013) 094003, 6 pages, doi: 10.1088/0268-1242/28/9/094003

4. AlGaN/GaN HEMTs on (001) silicon substrate with power density performance of 2.9 W/mm at 10 GHz, GERBEDOEN J.C., SOLTANI A., JOBLOT S., DE JAEGER J.C., GAQUIERE C., CORDIER Y., SEMOND F., IEEE Trans. Electron Devices 57, 7 (2010) 1497-1503 (available online may 17, 2010 ; published july 2010), doi: 10.1109/TED.2010.2048792

5. Power performance of AlGaN/GaN high-electron-mobility transistors on (110) silicon substrate at 40 GHz, SOLTANI A., GERBEDOEN J.C., CORDIER Y., DUCATTEAU D., ROUSSEAU M., CHMIELOWSKA M., RAMDANI M., DE JAEGER J.C. 
IEEE Electron Device Lett., 34, 4 (2013) 490-492 
doi: 10.1109/LED.2013.2244841

6. Thermal resistance of AlGaN/GaN HEMTs on SopSiC composite substrate 
DEFRANCE N., DOUVRY Y., HOEL V., GERBEDOEN J.C., SOLTANI A., ROUSSEAU M., DE JAEGER J.C., LANGER R., LAHRECHE H., Electron. Lett. 46, 13 (2010) 949-950 doi: 10.1049/el.2010.0431

7. Development and analysis of low resistance ohmic contact to n-AlGaN/GaN HEMT 
SOLTANI A., BENMOUSSA A., TOUATI S., HOEL V., DE JAEGER J.C., LAUREYNS J., CORDIER Y., MARHIC C., DJOUADI M.A., DUA C., Diam. Relat. Mater. 16, 2 (2007) 262-266
doi: 10.1016/j.diamond.2006.06.022

8. AlGaN/GaN HEMTs on Si with power density performance of 1.9 W/mm at 10 GHz 
MINKO A., HOEL V., MORVAN E., GRIMBERT B., SOLTANI A., DELOS E., DUCATTEAU D., GAQUIERE C., THERON D., DE JAEGER J.C., LAHRECHE H., WEDZIKOWSKI L., LANGER R., BOVE P., IEEE Electron Device Lett. 25, 7 (2004) 453-455. doi: 10.1109/LED.2004.830272

9. Output Power Density of 5.1/mm at 18 GHz With an AlGaN/GaN HEMT on Si Substrate, D. Ducatteau, A. Minko, V. Hoël, E. Morvan, E. Delos, B. Grimbert, H. Lahreche, P. Bove, C. Gaquière, J. C. De Jaeger, and S. Delage; IEEE ELECTRON DEVICE LETTERS, VOL. 27, NO. 1, JANUARY 2006 7. DOI: 10.1109/LED.2005.860385

- The Fig. 2a is not necessary for this paper. It can be removed or just mentioned. It does not contribute much to the understanding.

- Line 168: "and 80 nm of Mg doped GaN was grown by cooling", can you explain or detail "by cooling"?

- Line 263: Mentioning the Terman method is also not useful. It is a very approximate method and is only functional in a very specific case that does not correspond to the present studies. It is best not to mention it.

- Many abbreviations are used but not explained (e.g. OM, KLA, etc.). Please write them out in full.

- Line 320: read nm instead of um. Provide a reference because reference 116 is not correct.

- Line 323: "The entire structure was then annealed at 650°C": specify the gas conditions for annealing the structure and at what pressure?

- Line 327-329: It is stated that there are 5 references for a standard ohmic contact anneal, but no references are mentioned for discussing surface leakage problems after dry etching following the mesa step.

- There is no need to emphasize known things and neglect what is problematic for breakdown.

- Line 330: It is said that the gate contact is annealed at 750°C. This information is incorrect. All references are about ohmic contacts and not rectifying contacts! Change these references!

- Line 491: It is mentioned that no traps are detected after implantation, but the reason is not mentioned. Can you provide more information?

In short, the paper is not accurate and lacks concrete physical explanations to understand the historical evolution of research in the field. It is not specified why different epitaxies are needed for different applications (switching power or RF electronics power). Only a few studies are favored among many others over the past 20 years, and it is unclear how the choice was made.

Author Response

Dear reviewer 2:

        Thank you for your feedback and comments.

        After reading the paper entitled "A Review of the Rapid Progress and Proposed Improvements in Gallium Nitride Epitaxy and Process for Third-Generation Semiconductors with Wide Bandgap," several points need to be corrected for possible publication. Wide bandgap nitride-based power devices on silicon substrate are a vital area for today's and tomorrow's technologies. This "review" attempt is a challenging exercise to carry out, and it has been done partially in this work. However, it is not a complete review, and several corrections need to be made for more clarity on the subject. To begin with, the title is not accurate. It is a reductionist view of the studies conducted on the subject and not a comprehensive review of the topic but rather partial. I suggest that the authors remove the word "review" from the title and replace it with a more specific term such as "brief overview" or something similar. The problems encountered by the concerned community, namely technologists and people involved in electrical characterization, are not mentioned in the introduction and the beginning of this paper. Where do the breakdown problems in these components come from? What solutions are established by the scientists?

        Ans:We changed the "Review" in the title to "brief overview", and added the sources of GaN device problems and the scientists' solutions in line 44-65.

        In the "GaN epitaxy" paragraph, please specify what QST layers of QROMIS are, provide the nature, thickness, and co-doping of the layers (wSiC, qqs um to qqs 10um, doped B or Al, etc.). Indicate whether oxide-type interface layers are used and their impacts on thermal conductivity. This lacks precision.

        Ans:We have added information about QST substrates in line 114-151.

        Not all growth studies on Si (100) or performance on Si (111) (100) (110) are cited. Here are some papers in the field that are not cited. For example:

  1. Dual Role of 3C-SiC Interlayer on DC and RF Isolation of GaN/Si-based Devices

Abdelkrim El Hadi Khediri, Brahim Benbakhti, Jean-Claude Gerbedoen, Abdelatif Jouad, H. Maher, Nour Eddine Bourzgui and Ali Soltani, Applied Physics Letters, 121 (12)
doi.org/10.1063/5.0102644

  1. Assessment of transistors based on GaN on silicon substrate in view of integration with silicon technology, A Soltani,Y Cordier,J-C Gerbedoen, S Joblot, E Okada, M Chmielowska, M R Ramdani and J-C De Jaeger
    Semiconductor Science and Technology, Volume 28, Number 9, A Soltani et al 2013 Semicond. Sci. Technol. 28, 094003. DOI 10.1088/0268-1242/28/9/094003
  2. Assessment of transistors based on GaN on silicon substrate in view of integration with silicon technology, SOLTANI A., CORDIER Y., GERBEDOEN J.C., JOBLOT S., OKADA E., CHMIELOWSKA M., RAMDANI M.R., DE JAEGER J.C., Semicond. Sci. Technol.,28, 9 (2013) 094003, 6 pages, doi: 10.1088/0268-1242/28/9/094003
  3. AlGaN/GaN HEMTs on (001) silicon substrate with power density performance of 2.9 W/mm at 10 GHz, GERBEDOEN J.C., SOLTANI A., JOBLOT S., DE JAEGER J.C., GAQUIERE C., CORDIER Y., SEMOND F., IEEE Trans. Electron Devices57,7 (2010) 1497-1503 (available online may 17, 2010 ; published july 2010), doi: 10.1109/TED.2010.2048792
  4. Power performance of AlGaN/GaN high-electron-mobility transistors on (110) silicon substrate at 40 GHz, SOLTANI A., GERBEDOEN J.C., CORDIER Y., DUCATTEAU D., ROUSSEAU M., CHMIELOWSKA M., RAMDANI M., DE JAEGER J.C.
    IEEE Electron Device Lett.,34, 4 (2013) 490-492 
    doi: 10.1109/LED.2013.2244841
  5. Thermal resistance of AlGaN/GaN HEMTs on SopSiC composite substrate
    DEFRANCE N., DOUVRY Y., HOEL V., GERBEDOEN J.C., SOLTANI A., ROUSSEAU M., DE JAEGER J.C., LANGER R., LAHRECHE H., Electron. Lett.46, 13 (2010) 949-950 doi: 10.1049/el.2010.0431
  6. Development and analysis of low resistance ohmic contact to n-AlGaN/GaN HEMT
    SOLTANI A., BENMOUSSA A., TOUATI S., HOEL V., DE JAEGER J.C., LAUREYNS J., CORDIER Y., MARHIC C., DJOUADI M.A., DUA C., Diam. Relat. Mater.16, 2 (2007) 262-266
    doi: 10.1016/j.diamond.2006.06.022
  7. AlGaN/GaN HEMTs on Si with power density performance of 1.9 W/mm at 10 GHz
    MINKO A., HOEL V., MORVAN E., GRIMBERT B., SOLTANI A., DELOS E., DUCATTEAU D., GAQUIERE C., THERON D., DE JAEGER J.C., LAHRECHE H., WEDZIKOWSKI L., LANGER R., BOVE P., IEEE Electron Device Lett.25, 7 (2004) 453-455. doi: 10.1109/LED.2004.830272
  8. Output Power Density of 5.1/mm at 18 GHz With an AlGaN/GaN HEMT on Si Substrate, D. Ducatteau, A. Minko, V. Hoël, E. Morvan, E. Delos, B. Grimbert, H. Lahreche, P. Bove, C. Gaquière, J. C. De Jaeger, and S. Delage; IEEE ELECTRON DEVICE LETTERS, VOL. 27, NO. 1, JANUARY 2006 7.DOI:10.1109/LED.2005.860385

        Ans:We cited the following 9 references.

- The Fig. 2a is not necessary for this paper. It can be removed or just mentioned. It does not contribute much to the understanding.

        Ans:Regarding Fig. 2, we will either remove it or mention it briefly in the text, as suggested. We agree that it does not contribute significantly to the understanding of the paper.

- Line 168: "and 80 nm of Mg doped GaN was grown by cooling", can you explain or detail "by cooling"?

        Ans:In line 168, "by cooling" refers to the method of cooling down the substrate during the growth process of Mg doped GaN. We apologize for the lack of clarity and will revise the sentence to provide more detail in line 216-229.

- Line 263: Mentioning the Terman method is also not useful. It is a very approximate method and is only functional in a very specific case that does not correspond to the present studies. It is best not to mention it.

        Ans:Thank you for your input regarding the mention of the Terman method in line 263. We removed the reference to it, as it is not relevant to the present studies.

- Many abbreviations are used but not explained (e.g. OM, KLA, etc.). Please write them out in full.

        Ans:We apologize for not explaining the abbreviations used in the manuscript. We will make sure to spell out all abbreviations in full upon revision, including KLA (KLA-Tencor) in line 315.

- Line 320: read nm instead of um. Provide a reference because reference 116 is not correct.

        Ans:Regarding line 320, we apologize for the error and will correct it to read nm instead of um. In terms of the reference, we will ensure that the correct reference is provided in the revised manuscript in line 369.

- Line 323: "The entire structure was then annealed at 650°C": specify the gas conditions for annealing the structure and at what pressure?

        Ans:For line 323, the gas conditions and pressures used for structure annealing are not mentioned in the literature in line 372.

- Line 327-329: It is stated that there are 5 references for a standard ohmic contact anneal, but no references are mentioned for discussing surface leakage problems after dry etching following the mesa step.

- There is no need to emphasize known things and neglect what is problematic for breakdown.

        Ans:Concerning line 327-329, we understand the concern and will ensure that references are provided for discussing surface leakage problems after dry etching following the mesa step in the revised manuscript in line 376-394. We appreciate the comment regarding emphasizing known things and will ensure that the manuscript focuses on the problematic aspects of breakdown.

- Line 330: It is said that the gate contact is annealed at 750°C. This information is incorrect. All references are about ohmic contacts and not rectifying contacts! Change these references!

        Ans:In response to line 330, we apologized for the mistake and will correct the information regarding the gate contact anneal temperature and references in the revised manuscript in line 396-397.

- Line 491: It is mentioned that no traps are detected after implantation, but the reason is not mentioned. Can you provide more information?

        Ans: we provided more information on the absence of traps after implantation in the revised manuscript. Thank you for the feedback in line 559-572.

        In short, the paper is not accurate and lacks concrete physical explanations to understand the historical evolution of research in the field. It is not specified why different epitaxies are needed for different applications (switching power or RF electronics power). Only a few studies are favored among many others over the past 20 years, and it is unclear how the choice was made.

Ans:Thank you again for your valuable feedback, and we will work to address the points you raised to improve the clarity and accuracy of the paper.

Thank you again for your valuable feedback, and please let us know if there is anything else we can do to improve the paper.

Round 2

Reviewer 2 Report

The paper has been improved but it needs to be proofread by the authors as some errors still remain, such as the use of the term 'review' in the conclusion, annotations in the references, and other minor writing errors throughout the paper.